# REVEALING TASK-DEPENDENT LAYER RELEVANCE VIA ATTENTIVE MULTI-LAYER FUSION

**Marco Morik**[*][†]
Machine Learning Group
TU Berlin
BIFOLD [‡]

**Laure Ciernik**[*][†]
Machine Learning Group
TU Berlin
Hector Fellow Academy
ELLIS

**Lukas Thede**
University of Tübingen
Tübingen AI Center
Helmholtz Munich
MCML [§]

**Luca Eyring**
Helmholtz Munich
TU Munich
MCML [§]

**Shinichi Nakajima**
Machine Learning Group
TU Berlin
BIFOLD [‡]
RIKEN AIP

**Zeynep Akata**
Helmholtz Munich
TU Munich
MCML [§]

**Lukas Muttenthaler** [†]
Aignostics GmbH
Helmholtz Munich
TU Munich
MCML [§]

## ABSTRACT

Efficiently adapting large-scale foundation models to downstream tasks is a central challenge in modern deep learning. While linear probing is a standard and computationally efficient method, it typically operates exclusively on the final layer's representation. In this work, we present experimental evidence that this approach discards crucial task-relevant information distributed across other layers of the network. To investigate this, we introduce Attentive Layer Fusion (ALF), a probing mechanism that dynamically fuses representations from all layers of Vision Transformers. Acting as a an investigation tool, ALF reveals that optimal representation depth is highly task-dependent: while tasks similar to the pre-training domain rely on the final layer, specialized domains (e.g., medical, satellite) benefit significantly from intermediate layers. Furthermore, by analyzing representational similarities, we show that intermediate layers often achieve high downstream performance despite having low similarity to the final layer, indicating they encode distinct, complementary features. Across 19 diverse datasets and 9 foundation models, our hierarchical approach achieves consistent gains, offering a new lens into how foundation models organize information.

## 1 INTRODUCTION

Foundation models (Radford et al., 2021; Oquab et al., 2024) are trained to build hierarchical abstractions of their input. It is generally understood that early layers capture low-level structural cues (edges, textures), while later layers encode high-level semantic concepts aligned with the pre-training objective (Raghu et al., 2021; Dorszewski et al., 2025). However, linear probing, the standard methodology for adapting these models to downstream tasks, typically uses only the final layer's `CLS` token.

This design implicitly assumes that the final layer is a sufficient statistic for all downstream tasks. While recent work has begun to challenge this assumption by concatenating late layers (Oquab et al., 2024) or using features across the network's hierarchy (Tu et al., 2023; Wu et al., 2024; Bolya et al., 2025), a systematic understanding about the specific location of task-relevant information within a model is missing. We hypothesize that as a model progresses through its depth, it compresses information to satisfy its pre-training objective, potentially "abstracting away" structural or textural details.

---

[*]Equal contribution.

[†]Correspondence to *m.morik@tu-berlin.de,ciernik@tu-berlin.de*, or *lukas.muttenthaler@tu-berlin.de*.

[‡]Berlin Institute for the Foundations of Learning and Data, Berlin, Germany.

[§]Munich Center for Machine Learning, Munich, Germany.

In this work, we propose **Attentive Layer Fusion (ALF)** as a probe to analyze the distribution of information within Vision Transformers (ViTs). ALF uses an attention probe Chen et al. (2024) that attends to summary tokens (both average-pooled AP and CLS) from *all* layers simultaneously, automatically discovering the most relevant abstraction level for a given task. Our experiments across 19 datasets reveal distinct empirical regularities. We find that intermediate layers are often superior to the final layer for tasks that differ from the pre-training domain. We use Centered Kernel Alignment (CKA)(Kornblith et al., 2019a) to show that layers with low similarity to the final output can still drive high performance. Additionally, attention heatmaps extracted from our probe reveal dataset-dependent patterns. For natural images, the model relies more on later layers, while for structured or specialized datasets, the intermediate layers appear more crucial. These findings suggest that future adaptation methods must be depth-aware, using the entire backbone as an input rather than just the final output tokens.

## 2 METHOD

To probe the hierarchy, we extract representations from all $\mathcal{L}$ layers of a frozen ViT. For each layer $\ell$, we extract two complementary summary tokens: the global classification token $\boldsymbol{h}^{(\ell)}_{\text{[CLS]}}$ and the spatial average $\boldsymbol{h}^{(\ell)}_{\text{[AP]}}$ (Average Pool).

We include $\boldsymbol{h}^{(\ell)}_{\text{[AP]}}$ because the [CLS] token in early layers is often not fully contextualized, whereas spatial averaging captures low-level statistics and texture information that may be preserved throughout the hierarchy. We stack these to form a memory bank of hierarchical features $\boldsymbol{H}_{\mathcal{L}} \in \mathbb{R}^{2L \times d}$.

We employ a multi-head cross-attention mechanism to fuse these features Chen et al. (2024). A learnable query vector $\boldsymbol{Q}$ (acting as a "task prototype") attends to the layer representations. For each head $m$, we compute:

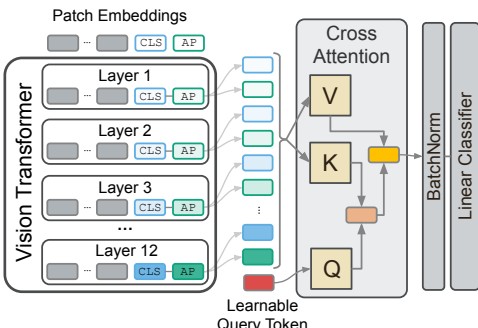

**Attentive Probe CLS + AP tokens of** $\mathcal{L}$

Figure 1: **Attentive Layer Fusion (ALF).** By treating layers as a sequence to be queried, ALF automatically discovers the most relevant abstraction level for a given task.

$$\text{Attention}(\boldsymbol{Q}^{(m)}, \boldsymbol{H}_{\mathcal{L}}) = \text{softmax}\left(\frac{\boldsymbol{Q}^{(m)}(\boldsymbol{H}_{\mathcal{L}}\boldsymbol{W}_k^{(m)})^{\top}}{\sqrt{d_h}}\right)(\boldsymbol{H}_{\mathcal{L}}\boldsymbol{W}_v^{(m)}) \tag{1}$$

The attention weights produced by the softmax provide a direct, interpretable measure of **layer relevance**. If the model assigns a high weight to Layer $k$, we infer that Layer $k$ contains information critical for the task that is either lost or obfuscated in other layers. An illustration of our approach can be seen in Fig. 1.

## 3 EXPERIMENTS

We evaluate our hypothesis on 19 datasets from VTAB and the CLIP-benchmark, covering natural, specialized, and structured domains. We utilize 9 foundation models from three families: CLIP (image-text aligned), DINOv2 (self-supervised), and supervised ViTs, spanning Small, Base, and Large sizes. Full experimental details are provided in Section A and the code is available under `https://github.com/lciernik/attentive-layer-fusion`.

### 3.1 INTERMEDIATE LAYERS CONTAIN DISTINCT, NON-REDUNDANT FEATURES

To understand the nature of the information distributed across the hierarchy, we first examine the relationship between downstream performance and representational similarity. We train linear probes on each layer individually and compute the CKA similarity with an RBF kernel ($\sigma = 0.2$) (Kornblith et al., 2019a) between each intermediate layer and the final layer representation.

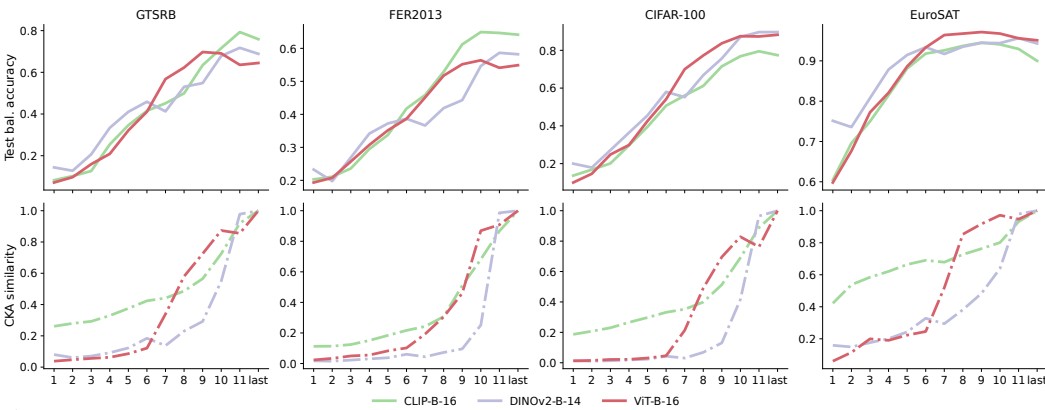

Figure 2: **Downstream performance vs. Representational Similarity. Top:** Balanced accuracy of linear probes trained on individual layers. **Bottom:** CKA similarity between Layer $i$ and the Final Layer. On specialized tasks like GTSRB or EuroSAT, performance often peaks in middle layers despite these layers having very low similarity to the final representation. This indicates that vital task information is discarded in the final layers.

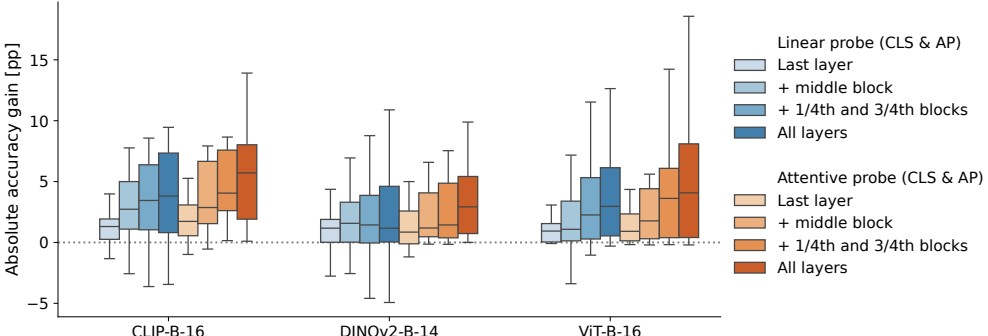

Figure 3: **Performance gain over standard Linear Probing.** Adding intermediate layers consistently improves performance across CLIP, DINOv2, and ViT backbones. ALF (orange) effectively selects relevant information, whereas naive linear concatenation (blue) suffers from high variance and instability.

Fig. 2 reveals a disconnect between semantic proximity and task utility. As shown in the bottom row, the CKA similarity to the final layer remains low throughout the network, rising sharply only in the final 2-3 layers. If the final layer were a comprehensive summary, we would expect accuracy to follow a similar trend. However, the top row shows that downstream accuracy often rises much earlier and, crucially, frequently peaks in the middle of the network.

This phenomenon is most visible on out-of-distribution tasks. For example, on EuroSAT (satellite imagery), the DINOv2 backbone achieves high accuracy around layers 6-8, where its CKA similarity to the final layer is negligible ($< 0.3$). As the model progresses to layer 12, similarity to the final state naturally maximizes, but downstream accuracy often stagnates. Two explanations could account for this behavior. Either the final layer is primarily a refinement of earlier representations, in which case intermediate layers add little beyond redundancy. Alternatively, the network reallocates information across depth, trading structural cues for more abstract semantics to satisfy its pre-training objective. In that case intermediate layers retain signals not preserved in the final representation.

## 3.2 INFORMATION IS DISTRIBUTED HIERARCHICALLY

We next quantify the benefit of accessing this distributed information. We measure the absolute accuracy gain relative to a standard linear probe on the final [CLS] token. To separate the benefit of information availability (access to the layers) from information selection (which layers to use), we compare two fusion strategies: naive Linear Concatenation (blue) and Attentive Layer Fusion (ALF, orange). For both, we vary the input scope: accessing only the last layer, adding the middle layer, adding quarterly intervals, and finally accessing the complete hierarchy.

As shown in Fig. 3, performance scales positively with the inclusion of intermediate layers. Merely adding the middle layer improves accuracy over the baseline, and accessing the full hierarchy yields the highest median gain.

Crucially, these results highlight the distinction between availability and accessibility. While naive linear concatenation of all layers improves performance on average, it suffers from high variance,

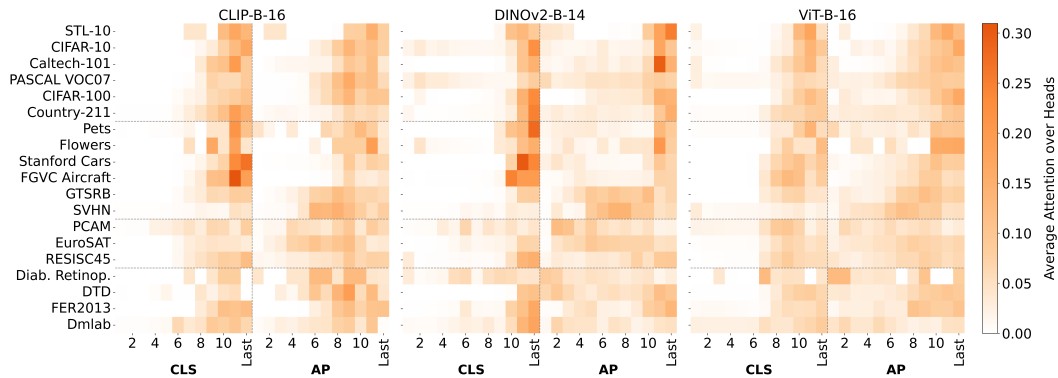

Figure 4: **Layer Relevance Heatmaps.** We visualize the learned attention weights for different datasets on Base models. **Left:** Natural image tasks (CIFAR, Pets) focus heavily on the final layers (11-12). **Right:** Out-of-distribution tasks (EuroSAT, textures, medical) shift attention to intermediate layers (5-9), indicating that the final layer has abstracted away necessary structural information.

leading to performance degradation on some tasks due to the curse of dimensionality. In contrast, ALF acts as a selective filter. By dynamically downweighting irrelevant layers, it achieves consistent gains ($\approx 5.5$ percentage points on average) without performance deficits. This confirms that while task-relevant information is distributed throughout the hierarchy, it must be carefully selected.

### 3.3 DOMAIN SHIFT DICTATES LAYER RELEVANCE

Finally, we visualize how ALF utilizes this distributed information. Fig. 4 plots the learned attention weights, acting effectively as a "depth-meter" for domain shift.

The heatmaps reveal two distinct behaviors depending on the downstream task. For **in-distribution (Natural Images)** datasets such as CIFAR-10, Oxford Pets, or Flowers102, the model attends primarily to the final layers, often prioritizing the CLS token. This validates that for tasks semantically aligned with the pre-training data, the standard assumption, that the final layer contains the optimal representation, holds true.

In contrast, for **out-of-distribution (Specialized/Structured)** datasets involving textures (DTD), satellite imagery (EuroSAT, Resisc45), or medical data (PCAM), the center of gravity shifts significantly towards intermediate layers and the spatial AP token. In the case of EuroSAT, the model explicitly retrieves information from the AP tokens of layers 6-9. This aligns directly with our earlier CKA analysis (Section 3.1), confirming that the model "reaches back" to retrieve structural features that were transformed or discarded in the final semantic abstraction. Overall, these heatmaps demonstrate that ALF serves not just as an adaptation method but also as an accessible diagnostic tool for analyzing the layer-wise utility of foundation models for any given task.

## 4 CONCLUSION AND DISCUSSION

In this paper, we presented Attentive Layer Fusion (ALF), a method that leverages the full depth of Vision Transformers to improve downstream adaptation. Beyond performance gains, our work serves as an investigation tool into the internal structure of foundation models, challenging the widespread view that the final layer is the only source of task-relevant information (Raghu et al., 2021; Kornblith et al., 2019b). We provide empirical evidence that the last layer is often an arbitrary truncation point. For specialized domains like medical or satellite imagery, the most valuable signals are effectively "left behind" in the middle of the network.

Crucially, we demonstrate that accessing this distributed information requires more than simple concatenation. While a naive linear combination of layers leads to instability and overfitting, our attentive mechanism acts as a selective filter, reliably identifying the optimal level of abstraction for each task. These observations align with similar findings in Language Models (Skean et al., 2025), suggesting a fundamental principle for foundation models across modalities: as models compress information to satisfy their pre-training objectives, they abstract away details that are crucial for specialized downstream tasks. Consequently, future research, whether in vision, language, or emerging biological models (Brixi et al., 2025), should view adaptation not as a simple mapping from the final output, but as a search for task-relevant information across all layers of a model.

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

# A  IMPLEMENTATION DETAILS

Table 1: Overview of the 19 datasets used in our experiments including the size of both train and test set, number of classes, and the Class Imbalance Ratio (CIR) calculated by $\frac{N_{\text{Majority Class}}}{N_{\text{Minority Class}}}$.

| Category | Dataset | Train Size | Test Size | Classes | CIR | Reference |
|---|---|---|---|---|---|---|
| Natural (MD) | STL-10 | 5 000 | 8 000 | 10 | 1 | Coates et al. (2011) |
| | CIFAR-10 | 45 000 | 10 000 | 10 | 1.02 | Krizhevsky (2009) |
| | Caltech-101 | 2 753 | 6 085 | 102 | 1.3 | Fei-Fei et al. (2006) |
| | PASCAL VOC 2007 | 7 844 | 14 976 | 20 | 20.65 | Everingham et al. (2010) |
| | CIFAR-100 | 45 000 | 10 000 | 100 | 1.06 | Krizhevsky (2009) |
| | Country-211 | 31 650 | 21 100 | 211 | 1 | Radford et al. (2021) |
| Natural (SD) | Pets | 2 944 | 3 669 | 37 | 1.24 | Parkhi et al. (2012) |
| | Flowers | 1 020 | 6 149 | 102 | 1 | Nilsback & Zisserman (2008) |
| | Stanford Cars | 8 144 | 8 041 | 196 | 2.83 | Krause et al. (2013) |
| | FGVC Aircraft | 3 334 | 3 333 | 100 | 1.03 | Maji et al. (2013) |
| | GTSRB | 26 640 | 12 630 | 43 | 10 | Stallkamp et al. (2012) |
| | SVHN | 65 931 | 26 032 | 10 | 2.98 | Netzer et al. (2011) |
| Specialized | PCAM | 262 144 | 32 768 | 2 | 1 | Veeling et al. (2018) |
| | EuroSAT | 16 200 | 5 400 | 10 | 1.58 | Helber et al. (2019) |
| | RESISC45 | 18 900 | 6 300 | 45 | 1.16 | Cheng et al. (2017) |
| | Diabetic Retinopathy | 35 126 | 42 670 | 5 | 36.45 | Dugas et al. (2015) |
| Structured | DTD | 1 880 | 1 880 | 47 | 1 | Cimpoi et al. (2014) |
| | FER2013 | 28 709 | 7 178 | 7 | 16.55 | Goodfellow et al. (2015) |
| | Dmlab | 65 550 | 22 735 | 6 | 1.98 | Zhai et al. (2020) |

**Feature Extraction & Preprocessing.** We extract representations using `thingsvision` (Muttenthaler & Hebart, 2021). Images are resized to 256px and center-cropped to 224px. Extracted features are L2-normalized. We use a frozen backbone strategy, training only the fusion module and classifier. To handle class imbalance, we employ a weighted cross-entropy loss where class weights are inversely proportional to class frequency (Aurelio et al., 2019).

**Training & Optimization.** Models are trained for $\geq 40$ epochs (ensuring $\geq 1000$ steps) using AdamW with cosine annealing. We use a batch size of up to 2048. To prevent overfitting, we apply gradient clipping (norm 5.0), weight decay, and inject Gaussian noise ($\mathcal{N}(0, 0.05)$) to representations during training. Hyperparameters were selected via grid search on a validation split (80/20): Learning rates $\in \{10^{-1}, \ldots, 10^{-3}\}$, Dropout $\in \{0.0, 0.1, 0.3\}$, Weight decay $\in \{10^{-6}, \ldots, 1.0\}$.

**Attention Mechanism.** We base the attentive probe on the modle of Chen et al. (2024). For ALF, the number of attention heads $M$ matches the number of fused representations (e.g., $M = 24$ for 12 layers of CLS + AP). Queries are initialized from $\mathcal{N}(0, 0.02)$. Code is based on Ciernik et al. (2025) and will be released.

**Evaluation Metric** To enable an intuitive comparison of performances across datasets, we report the absolute top-1 accuracy gain (in percentage points [pp]) of each method over the standard linear probe CLS baseline: $\text{Acc}_{\text{bal}}(\text{method}) - \text{Acc}_{\text{bal}}(\text{CLS}_{\text{linear}})$, which is positive if the method outperforms the baseline.

**Model Details** We evaluate three model families across Small, Base, and Large scales:

- **Supervised ViT:** ViT-S/16, ViT-B/16, and ViT-L/16 pretrained on ImageNet-21K and fine-tuned on ImageNet-1K (Deng et al., 2009; Ridnik et al., 2021).
- **Self-Supervised DINOv2:** ViT-S-14, ViT-B-14, and ViT-L-14 , pretrained on the LVD-142M dataset (Oquab et al., 2024).
- **Image-Text Alignment CLIP:** OpenCLIP models ViT-B-32, ViT-B-16, and ViT-L-14 (Cherti et al., 2023; Ilharco et al., 2021)) following the CLIP architecture and using its pretrained weights (Radford et al., 2021)).

**Dataset Details** An overview of all datasets used in this work is given in Tab. 1. Following VTAB (Zhai et al., 2020), the datasets are categorized by domain.

