# OpenReview forum: "Revealing Task-Dependent Layer Relevance via Attentive Multi-Layer Fusion"
_ICLR.cc/2026/Workshop/Sci4DL — Sci4DL 2026_

### Official Review · Reviewer_BLLx · 2026-02-24

**Fit:** 2
**Significance:** 2
**Confidence:** 3

**Summary:**

This paper challenges the common practice of linear probing only the final layer of Vision Transformers. The authors propose Attentive Layer Fusion (ALF), a cross-attention-based probing mechanism that dynamically fuses CLS and average-pooled (AP) tokens from all layers of a frozen backbone. Across 19 datasets and 9 foundation models (CLIP, DINOv2, supervised ViTs), the authors show:
1. Intermediate layers often outperform the final layer on out-of-distribution tasks.
2. Representational similarity (CKA) to the final layer does not correlate with downstream utility.
3. Naive concatenation of layers improves performance but is unstable.
4. ALF yields consistent improvements.
5. Attention heatmaps reveal domain-dependent layer reliance.

**Strengths:**

1. The paper systematically challenges the common assumption that the final layer is sufficient for transfer, providing strong empirical evidence across 19 datasets and 9 foundation models that layer relevance is task-dependent.
2. The CKA vs. downstream performance analysis convincingly shows that intermediate layers can achieve high task accuracy despite low similarity to the final representation, demonstrating that they encode distinct, non-redundant features.
3. The ALF mechanism not only improves performance over naive layer concatenation but also provides interpretable attention heatmaps that reveal domain-dependent depth usage, making it both an adaptation method and a diagnostic tool.

**Suggestions:**

1. Strengthen the theoretical grounding by providing deeper analysis of why intermediate layers retain complementary information.
2. Include additional ablations and stability analysis, such as reporting variance across seeds, analyzing attention weight entropy/sparsity, and comparing ALF to simpler learned layer-weighting baselines to clarify what aspects of the method drive the gains.
3. Extend the evaluation beyond frozen linear probing, for example by testing whether the depth-aware effect persists under partial or full fine-tuning, which would clarify the broader applicability of the findings.

---

### Official Review · Reviewer_vVTT · 2026-02-25

**Fit:** 3
**Significance:** 3
**Confidence:** 2

**Summary:**

This paper challenges the conventional practice of relying solely on the final-layer representations for downstream task adaptation. The authors propose Attentive Layer Fusion, a probing mechanism that leverages a cross-attention module to dynamically fuse both CLS and global average pooled  tokens from all $L$ layers of a frozen Vision Transformer. Through extensive experiments across 19 datasets and 9 foundation models, the paper demonstrates that task relevance exhibits a strong depth dependency, highlighting the importance of adaptively aggregating representations from different layers rather than depending exclusively on the final layer.

**Strengths:**

1. The paper provides strong empirical evidence showing that the final layer is often an “arbitrary truncation point,” especially for domain specific tasks. This challenges the widespread assumption that the deepest representation is necessarily the most task relevant.

2. The methodology is rigorous and comprehensive, covering 19 diverse datasets and 9 models from three different families. This breadth of evaluation strengthens the generalizability of the conclusions.

3. By acting as a selective filtering mechanism, ALF outperforms simple linear concatenation. It avoids the curse of dimensionality and the training instability that commonly arise when naively stacking representations from all layers, demonstrating both conceptual elegance and practical effectiveness.

**Suggestions:**

This is a very strong paper, and therefore I have only one suggestion. Although the conclusions suggest that the findings may reflect a general principle applicable to cross modal models, the argument would be further strengthened by including experiments on other architectures—such as CNN-based models or large language models. Empirical validation beyond Vision Transformers would significantly reinforce the claim that this is indeed a “fundamental principle,” rather than a phenomenon specific to ViT-style architectures.

---

### Meta-Review · Area_Chair_ui1W · 2026-03-01

**Recommendation:** Accept

**Metareview:**

Both reviewers posted positive reviews. I recommend accept.

---

### Decision · Program_Chairs · 2026-03-02

Accept